# Personality reflection in the brain's intrinsic functional architecture remains elusive

**David Tomeček**[1,4,5], **Renata Androvičová**[1,6], **Iveta Fajnerová**[1], **Filip Děchtěrenko**[2], **Jan Rydlo**[1,3], **Jiří Horáček**[1], **Jiří Lukavský**[1,2], **Jaroslav Tintěra**[1,3], **Jaroslav Hlinka**[1,4]*

**1** National Institute of Mental Health, Klecany, Czech Republic, **2** Institute of Psychology, Czech Academy of Sciences, Prague, Czech Republic, **3** Department of Radiology, Institute for Clinical and Experimental Medicine, Prague, Czech Republic, **4** Institute of Computer Science, Czech Academy of Sciences, Prague, Czech Republic, **5** Faculty of Electrical Engineering, Czech Technical University in Prague, Prague, Czech Republic, **6** Third Faculty of Medicine, Charles University, Prague, Czech Republic

* hlinka@cs.cas.cz

## Abstract

In the last years, there has been a considerable increase of research into the neuroimaging correlates of inter-individual temperament and character variability—an endeavour for which the term 'personality neuroscience' was coined. Among other neuroimaging modalities and approaches, substantial work focuses on functional connectivity in resting state (rs-FC) functional magnetic resonance imaging data. In the current paper, we set out to independently query the questions asked in a highly cited study that reported a range of functional connectivity correlates of personality dimensions assessed by the widely used 'Big Five' Personality Inventory. Using a larger sample (84 subjects) and an equivalent data analysis pipeline, we obtained widely disagreeing results compared to the original study. Overall, the results were in line with the hypotheses of no relation between functional connectivity and personality, when more precise permutation-based multiple testing procedures were applied. The results demonstrate that as with other neuroimaging studies, great caution should be applied when interpreting the findings, among other reasons due to multiple testing problem involved at several levels in many neuroimaging studies. Of course, the current study results can not ultimately disprove the existence of some link between personality and brain's intrinsic functional architecture, but clearly shows that its form is very likely different and much more subtle and elusive than was previously reported.

## Introduction

In their lives, people encounter many different situations and they can act in many different ways. However, their behaviour is not random and it tends to be partly predictable. This invariance in how people think, feel and behave is being incorporated in the term personality [1]. Many concepts have been used to describe personality traits, but due to their mutual correlations, it is possible to describe human personality with a smaller number of underlying factors [2]. The popular solution is the Five-Factor Model or Big Five [3]. While this concept was

**Data Availability Statement:** The data required to replicate all study findings reported in the article, i.e. all the individual functional connectivity maps (i.e. the first-level beta maps) from each of the 18

seed regions for both of the applied preprocessing settings, as well as the covariate variables used (sex, age, and 5 personality dimension scores: neuroticism, extraversion, openness, agreeableness, conscientiousness)) are available at https://osf.io/69kj8/. The codes applied are provided in the article's Supporting Information.

**Funding:** This study was supported by the Czech Science Foundation (https://gacr.cz/en/) project No. 13-23940S (JHo, JL, JT, JHl) and project Nr. LO1611 with a financial support from the Ministry of Education, Youth and Sports (http://www.msmt.cz/?lang=2) under the NPU I program, and by the long-term strategic development financing of the Institute of Psychology (RVO: 68081740) and Institute of Computer Science (RVO:67985807) of the Czech Academy of Sciences. The funders had no role in study design, data collection and analysis, decision to publish, or preparation of the manuscript.

**Competing interests:** The authors have declared that no competing interests exist.

proposed initially as descriptive, there is increasing amount of literature linking it to the biological level.

In the recent years, there has been a considerable increase of interest in research into the neurobiological correlates of inter-individual behavioral differences. In this context, the term 'personality neuroscience' has been coined [4]. A wide range of measurement and data analysis methods have been used to find neuroimaging correlates of personality differences assessed by standard psychometric tools. For illustration, consider the selection of neuroimaging results related to individual dimensions of the Big Five model: Wei, Duan, Yang, Liao, Gao, Ding, et al. [5] found a linkage between the default mode network and neuroticism and extraversion. Neuroticism, as a personality trait which indexes the tendency to experience a negative effect, was associated with functioning of the amygdala and the prefrontal cortex [6, 7]. Also an evidence based on diffusion tensor imaging indicates positive correlation between neuroticism and measure of loss of white matter integrity in the anterior cingulum and tracts that connect the prefrontal cortex and the amygdala [8] but the breakdown in the white matter integrity could be more widespread [9]. Neurotic brain was argued to have overall less than optimal functional network organization and exhibits overall weaker functional connections [10]. Extraversion is characterized as a social dimension associated with a preference for seeking, engaging in social interactions, implication in gregariousness and excitement-seeking [11, 12]. Low frequency oscillations in the precuneus and the medial prefrontal cortex in the resting-state were found to have a relationship with a degree of extraversion [13]. Higher extraversion score also correlated with an increased amygdala resting-state functional connectivity with the putamen, temporal pole, insula, and occipital cortex [14] and the right precuneus and both superior and inferior parietal lobes [15]. A high degree of extraversion has been linked with a greater response to positive visual emotion cues in the amygdala [16]. Openness, sometimes described as intellect, is associated with imagination, intellectual engagement, and aesthetic interest was found to have relationship with functioning of the dorsolateral prefrontal cortex [17] and was also associated with increased activity in the right inferior parietal lobe and decreased activity in the bilateral superior parietal cortex and the left precuneus [15]. Agreeableness encompasses traits known as altruism, desires, rights of others, empathy and other forms [18–20] and was found to be positively correlated with the medial prefrontal cortex and anterior cingulate cortex. Conscientiousness relates to traits like orderliness and self-discipline [21] and was positively correlated with right superior parietal cortex [15].

Notably, several neuroimaging studies have attempted to find brain correlates of personality dimensions using whole brain analysis in search for association with any of measured personality dimensions. Among these, Adelstein, Shehzad, Mennes, DeYoung, Zuo, Kelly, et al. [22] reported that each domain of personality predicted resting state functional connectivity (rs-FC) of seed regions placed within the anterior cingulate and precuneus with a unique pattern of brain regions.

As part of a larger project exploring the neuroimaging correlates of personality, we have decided to mimic the analysis of Adelstein, Shehzad, Mennes, DeYoung, Zuo, Kelly, et al. [22] using an independent sample of data. Our main question was: does an independent study support the findings of Adelstein, Shehzad, Mennes, DeYoung, Zuo, Kelly, et al. [22]? In other words, do the five domains of neuroticism, extraversion, openness, agreeableness, conscientiousness according to NEO Five-Factor Inventory (NEO-FFI) [3, 23]—predict resting-state MRI functional connectivity as described by Adelstein, Shehzad, Mennes, DeYoung, Zuo, Kelly, et al. [22]?

We use a larger sample of 84 subjects (instead of 39 in the original study, that moreover used multiple scans from some of the participants), an equivalent data processing pipeline, and the Gaussian random field (GRF) approach for multiple testing correction which was used

in the original study. There were some differences in the data acquisition (see Discussion), however both schemes fall more or less within the standard resting state acquisition. To gain more insight into our results, we complemented our analysis by re-analysing our data with another (more conservative) preprocessing scheme, and also with using a permutation testing based inference instead of the Gaussian random field (GRF) approach for multiple testing correction.

## Materials and methods

### General design

We used similar study design and analytical approaches as were originally proposed by Adelstein, Shehzad, Mennes, DeYoung, Zuo, Kelly, et al. [22]. Additionally, in order to evaluate the results, two different scenarios of denoising of the functional MRI data and two different methods of statistical inference were used.

### Participants

We acquired MRI brain scans of 84 healthy controls (80 right-handed, 48 males, mean age 30.83 ± 8.48). The distribution of age and gender was similar to that in Adelstein, Shehzad, Mennes, DeYoung, Zuo, Kelly, et al. [22] (they reported results from 39 right-handed adults, including 18 males, with mean age 30 ± 8 years). All participants gave written informed consent. The study was approved by the Ethics Committee of IKEM (Institute for Clinical and Experimental Medicine in Prague, Czech Republic).

### Assessment (NEO-FFI)

The participants filled out a Czech version of NEO Five-Factor Inventory (NEO-FFI). The inventory consists of 60 items and it is used to assess five personality dimensions: 1) neuroticism, 2) extraversion, 3) openness, 4) agreeableness, 5) conscientiousness [3, 23, 24].

### Data acquisition

Data acquisition took place at the IKEM using Siemens TrioTim 3T MR machine. A high-resolution 3D anatomical T1-weighted image was acquired (TR = 2300 ms, TE = 4.63 ms, flip angle = 10˚, FOV 256 × 256, image matrix size 256 × 256, voxel size = 1 × 1 × 1 mm, 224 sagittal slices) using the magnetization prepared gradient echo (MPRAGE) sequence. Then, the functional T2*-weighted images with blood oxygenation level-dependent (BOLD) contrast, (TR = 2500 ms, TE = 30 ms, flip angle = 90˚, FOV 192 × 192, image matrix size 64 × 64, voxel size = 3 × 3 × 3 mm, 44 axial slices, 240 volumes in total for each subject) were collected using the echo-planar imaging (EPI) technique.

### ROI selection

We selected anterior cingulate cortex and precuneus as our two main areas of interest which were split into 18 spatially separated spherical seed regions of interest (ROIs) with diameter of 8 mm. The ROIs were placed as in the study by Adelstein, Shehzad, Mennes, DeYoung, Zuo, Kelly, et al. [22] and sample the key midline structures of the anterior cingulate cortex and the precuneus, two functionally heterogeneous brain areas involved in diverse aspects of cognition, that are commonly investigated in resting state functional connectivity studies. In particular, ten unilateral ROIs were placed in anterior cingulate cortex and eight unilateral ROIs in the precuneus. Complete list of the ROIs with their MNI coordinates is listed in the S1 Table.

## Functional data preprocessing

The preprocessing pipeline of the resting-state functional images was carried out using CONN toolbox (The Gabrieli Lab, McGovern Institute for Brain Research, MIT) in Matlab (The MathWorks, Inc.). The CONN toolbox used standard preprocessing modules from SPM8 toolbox (Wellcome Trust Centre for Neuroimaging, UCL). The preprocessing pipeline comprised of slice timing correction for continuous decreasing acquisition, motion correction which realigned all functional images to a mean functional image, normalization of the functional images into the MNI 152 standard space and spatial smoothing with 6 mm FWHM kernel.

## Nuisance signal regression

In our analysis we used two different denoising schemes in order to assess their potential impact on the final results.

The first approach corresponded to that of Adelstein, Shehzad, Mennes, DeYoung, Zuo, Kelly, et al. [22]. Prior to the denoising procedures in the CONN toolbox, all preprocessed functional images were mean-based intensity normalized by a factor of 10000 using the FSL's suit fslmaths command. Resulting time series were further band-pass filtered using the FFT-based filter with a frequency window of 0.009—0.1 Hz which suppresses the low-frequency fluctuations and physiological noise of higher frequency mostly generated by cardiac and respiratory function [25]. This was followed by quadratic detrending, which reduces trends in a time domain and also despiking, which reduces the influence of potential outlier scans. An average time series extracted from a whole brain, an average time series of the white matter, of the cerebrospinal fluid and as well as six motion parameters (calculated while performing realignment of the functional images—rotations and translations in all three cardinal directions X, Y, Z), were used in a linear regression to reduce their potential confounding effect. Final denoised time series of interest were further used in the first-level statistics.

The second denoising approach was based on a default denoising scheme, which is standardly implemented in the CONN toolbox (without a 'motion scrubbing' option). This approach comprised of the same band-pass filtering as mentioned above, using the FFT-based filter with 0.009—0.1 Hz frequency window. This was followed by a linear detrending of the time series. CompCor approach, which is implemented in the CONN toolbox, was used to perform a principal component analysis with time series corresponding to the white matter and the cerebrospinal fluid [26]. Five components of the white matter, five components of the cerebrospinal fluid and as well as six motion parameters with their 1st order temporal derivatives were used in a linear regression to reduce their confounding effect on the signal of interest. Final denoised time series of interest were further used in the first-level statistics.

## Statistical analysis

Similarly, as in the case of denoising, we have used two different methodologies also in the case of statistical inference, namely at the level of multiple testing correction procedure at the second-level (group-level) inference. In particular, the commonly used Gaussian random field theory-based approach used in the original paper is increasingly criticized as potentially giving rise to an alarming rate of false positive findings [27]. This motivated us to rerun the analyses using a permutation (randomization) testing approach that should be more robust in this respect.

Note that the multiple testing correction mentioned provides correction across the spatial domain (i.e. many thousands of brain voxels for which the functional connectivity to a particular seed is assessed). However, it does not provide correction across the multiple hypotheses

assessed (there are 18 seed regions considered for each of five personality domains and tests are carried out for both positive and negative effects). In line with the original article, we do not carry out any *explicit* correction across these 180 analyses, and the overall framework is thus prone to provide on average $180 \times 0.05 = 9$ false positive findings even in the case that there was no link between personality and brain functional connectivity. While the original study reported apparently a much higher number of observed relations than 9, a key question is whether this wealth of findings is reproducible in an independent analysis, or potentially an artifact of the applied Gaussian random field theory-based multiple testing correction across space in each of the 180 analyses.

A general linear model (a default method in the CONN to determine functional connectivity) was used to determine functional connectivity between average time series of each of the selected ROIs and that of every other voxel in a brain. Beta maps which were further used in second-level statistics in combination with the Gaussian random field theory were by default converted to Pearson's r correlation maps. The original first-level beta maps were also used as an input for second-level statistics based on permutation tests.

### The Gaussian random field theory

The first approach, along the lines of the original study, involved the Gaussian random field theory. Technically, the second-level statistics were conducted using the general linear model with all five personality domains for each subject controlled for both age and sex as covariates in the CONN toolbox. Resulting second-level t-maps were estimated for smoothness using the FSL's smoothest utility and further corrected for multiple comparisons with the Gaussian random field theory as implemented in the FSL's cluster utility. Final t-maps were thresholded at t>2.3 and p<0.05 (one-sided test), in line with Adelstein, Shehzad, Mennes, DeYoung, Zuo, Kelly, et al. [22]. This resulted in a total of 180 statistical maps, one for each of the defined ROIs, for each of the five personality domains, representing either a positive or negative relationship.

### Permutation tests

In the other approach, based on the permutation tests, we used individual first-level beta maps from each of the 18 ROIs, which we merged along all subjects in order to create 4D maps which were then used in the second-level inference. We used FSL's randomise tool for non-parametric permutation tests which enabled us to use a standard general linear model [28] with all five personality domains for each subject controlled for both age and sex as covariates. The tool was set to make 5000 permutations when creating the null distribution, and cluster-based threshold of t>2.3 was selected. Only results with p<0.05 were considered in a final assessment.

## Results

### Personality domain scores

The descriptive statistics of the five personality domain scores are shown below in Table 1.

### Functional connectivity correlates of personality domain scores

When using the original denoising scheme and statistical inference method (the Gaussian random field theory), we have observed significant results in 74 out of the 180 analyses carried out (see Table 2). These include a number of areas of functional connectivity correlates for each of

**Table 1. Descriptive statistics for the NEO-FFI Five personality domain scores.** The sample mean and standard deviation of scores for each personality domain.

| domain | mean | st.d. |
|---|---|---|
| neuroticism | -0.369 | ±1.117 |
| extraversion | 0.085 | ±0.858 |
| openness | 0.543 | ±1.075 |
| agreeableness | 0.284 | ±0.876 |
| conscientiousness | 0.274 | ±0.888 |

**Table 2. A number of analyses with statistically significant results when using the original denoising and the GRF approach.** Thresholded at t>2.3 and p<0.05 (corrected). Overall, 180 analyses were carried out, using 18 functional connectivity seeds for each personality domain and direction of effect.

| analysis | n | e | o | a | c | total |
|---|---|---|---|---|---|---|
| positive | 5 | 6 | 2 | 7 | 12 | 32 |
| negative | 6 | 5 | 5 | 10 | 16 | 42 |
| total | 11 | 11 | 7 | 17 | 28 | 74 |

the personality dimension. An average value of smoothness (estimated with FSL's smoothest utility) for the first-level statistical maps was 0.0361±0.0056, see S2 Table for particular values.

For a summary visualization of obtained results, see Fig 1. In general, we have observed widespread cortical and subcortical areas of significant relation of FC and personality. Visual comparison with Fig 2 of the original study by Adelstein, Shehzad, Mennes, DeYoung, Zuo, Kelly, et al. [22] suggest a rather weak overlap of the observed results. To obtain some quantitative evidence on the agreement between the results, we have computed the number of analyses in which both datasets provided at least one significant cluster (even if spatially distinct), as the

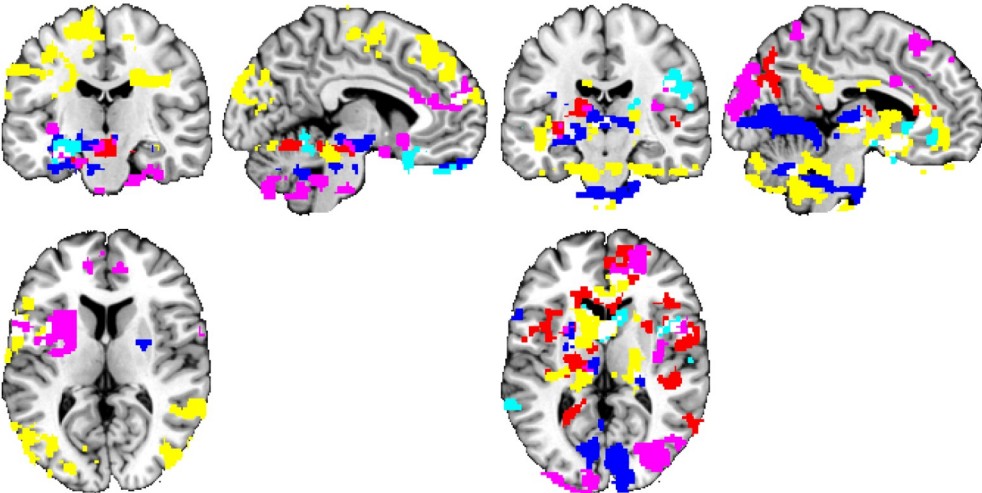

**Fig 1. Personality trait measures 'predicted' by rs-FC using the original denoising and the GRF approach.** Thresholded at t>2.3 and p<0.05 (corrected), positive—left, negative—right. Connections inferred as having a relationship with personality, grouped by color based on the personality domain: neuroticism = lightblue, extraversion = blue, openness = red, agreeableness = violet, conscientiousness = yellow. The significant functional connectivity maps of all 18 seeds are overlaid in a single image for compactness of presentation. Position of slices corresponds to MNI coordinates of -5,0,0.

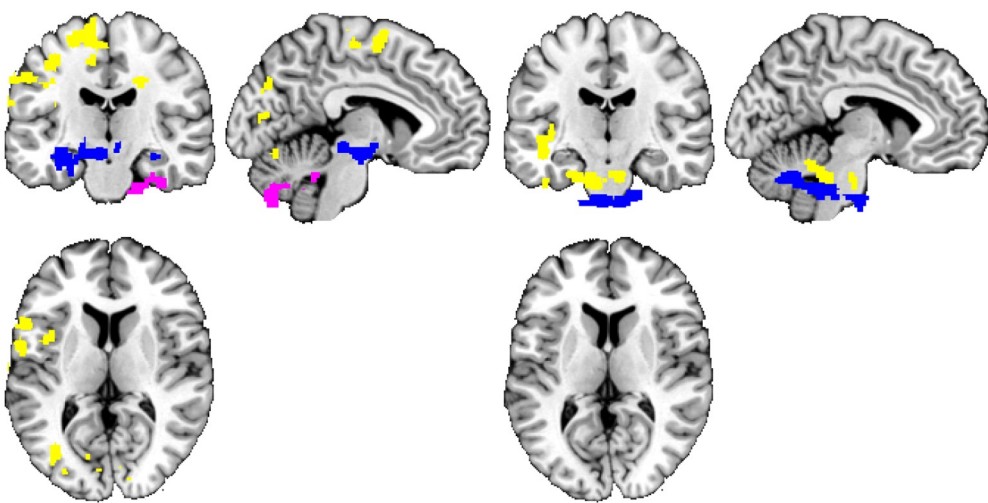

**Fig 2. Personality trait measures 'predicted' by rs-FC using the original denoising and the permutation-based approach.** Thresholded at t>2.3 and p<0.05 (corrected), positive—left, negative—right. Connections inferred as having a relationship with personality, grouped by color based on the personality domain: neuroticism = lightblue, extraversion = blue, openness = red, agreeableness = violet, conscientiousness = yellow. The significant functional connectivity maps of all 18 seeds are overlaid in a single image for compactness of presentation. Position of slices corresponds to MNI coordinates of -5,0,0.

exact spatial overlap is technically not possible to be determined based on the available results. The original study detected 106 of the 180 analyses as significant (using the Gaussian random field (GRF) approach for multiple testing correction), but only 42 of these were also among those 74 that were significant in our data.

The observed overlap between the two studies corresponds to result expected at random (the expected overlap count for randomly selected sets of hypotheses of the same cardinality is $180 \times (106 \div 180) \times (74 \div 180) = 43.58$ results). This only strengthens the suspicion, that the results of both studies amount to false positives. To gain more insight, we have repeated the analyses with a permutation-based inference scheme (instead of Gaussian random field theory) to control for multiple testing problems. Here, the extent of the results obtained was much smaller, see Fig 2. The resulting areas of personality-related functional connectivity clusters consisted generally of a spatially much more restricted subset of results of the initial analyses. Still, even the intersection of these two analyses, when merged across all 18 seeds and all personality domains and directions of change, provided a rich set of results.

In particular, significantly positive relationship with the resting-state seed-to-voxel functional connectivity was found for extraversion, agreeableness, and conscientiousness. Extraversion was found to have a significantly positive relationship with the temporal pole, the temporal fusiform cortex (posterior division), the parahippocampal gyrus, the insular cortex, and the planum polare. Agreeableness had a positive relationship with the functional connectivity in the frontal orbital cortex, the parahippocampal gyrus (anterior division), the subcallosal cortex, and the temporal pole. Conscientiousness was significantly positively correlated with the precentral gyrus, the lingual gyrus, the postcentral gyrus, the temporal occipital fusiform gyrus, the lateral occipital cortex (superior and inferior division), the central opercular cortex, the juxtapositional lobule cortex, the cuneal cortex, the temporal fusiform cortex (posterior division), and the inferior temporal cortex (temporooccipital part). Conversely, negative relationship between the seed-to-voxel functional connectivity of the default mode network and the other regions in the brain was found for conscientiousness in the parahippocampal

**Table 3. A number of analyses with statistically significant results when using the original denoising and the permutation-based approach.** Thresholded at t>2.3 and p<0.05 (corrected). Overall, 180 analyses were carried out, using 18 functional connectivity seeds for each personality domain and direction of effect.

| analysis | n | e | o | a | c | total |
|---|---|---|---|---|---|---|
| positive | 0 | 1 | 0 | 1 | 2 | 4 |
| negative | 0 | 1 | 0 | 0 | 1 | 2 |
| total | 0 | 2 | 0 | 1 | 3 | 6 |

gyrus (anterior division), the temporal pole, the temporal fusiform cortex (anterior division), the temporal occipital fusiform cortex, the temporal fusiform cortex (posterior division), the inferior temporal gyrus (anterior division), the planum polare, and the insular cortex.

Notably, in the case of permutation-based inference, we have observed only 6 of the 180 analyses provide a 'significant effect' result (see Table 3). Note that when controlling at p<0.05 FWE in each of the analyses, the expected number of significant analyses in a set of 180 is $180 \times 0.05 = 9$. In other words, the number of observed results from the permutation-based inference approach is in line with the hypothesis of no relation between personality and functional connectivity.

Using the alternative denoising scheme, we have observed quantitatively similar effect— namely a high number of distributed clusters of personality-related functional connectivity under the use of GRF-based statistical inference (an average value of smoothness (estimated with FSL's smoothest utility) for the first-level statistical maps was 0.0443 ± 0.0075, see S2 Table for particular values), falling down to a result consistent with the hypothesis of no relation, when permutation-based statistical inference was used. See Figs 3 and 4 and Tables 4 and 5 for reference.

Considering common results of both methods of statistical inference, using the default CONN denoising, openness exhibited a significantly positive relationship with the resting-

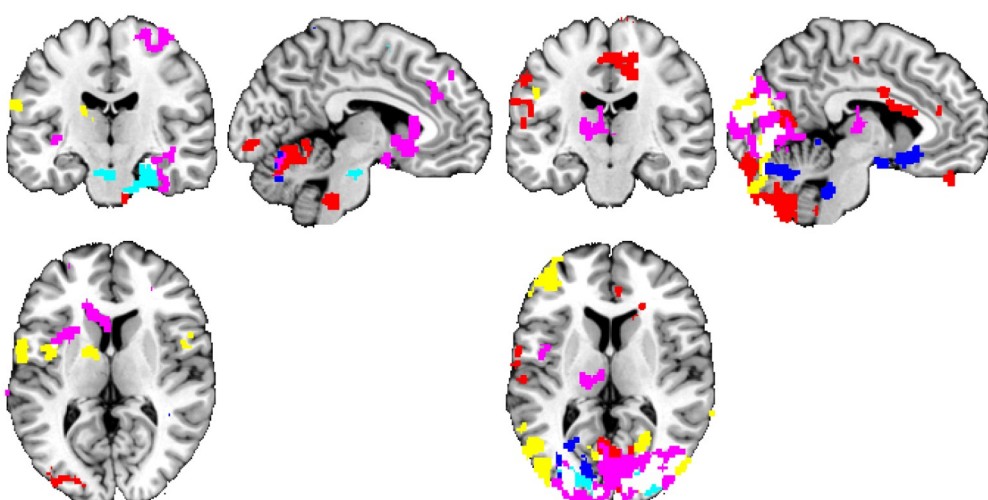

**Fig 3. Personality trait measures 'predicted' by rs-FC using the default CONN denoising and the GRF approach.** Thresholded at t>2.3 and p<0.05 (corrected), positive—left, negative—right. Connections inferred as having a relationship with personality, grouped by color based on the personality domain: neuroticism = lightblue, extraversion = blue, openness = red, agreeableness = violet, conscientiousness = yellow. The significant functional connectivity maps of all 18 seeds are overlaid in a single image for compactness of presentation. Position of slices corresponds to MNI coordinates of -5,0,0.

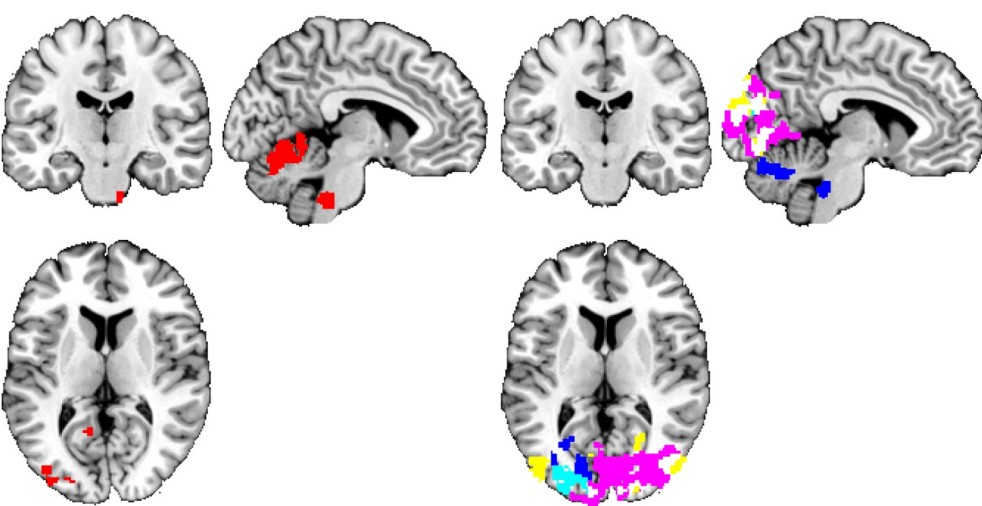

**Fig 4. Personality trait measures 'predicted' by rs-FC using the default CONN denoising and the permutation-based approach.** Thresholded at t>2.3 and p<0.05 (corrected), positive—left, negative—right. Connections inferred as having a relationship with personality, grouped by color based on the personality domain: neuroticism = lightblue, extraversion = blue, openness = red, agreeableness = violet, conscientiousness = yellow. The significant functional connectivity maps of all 18 seeds are overlaid in a single image for compactness of presentation. Position of slices corresponds to MNI coordinates of -5,0,0.

**Table 4. A number of analyses with statistically significant results when using the default CONN denoising and the GRF approach.** Thresholded at t>2.3 and p<0.05 (corrected). Overall, 180 analyses were carried out, using 18 functional connectivity seeds for each personality domain and direction of effect.

| analysis | n | e | o | a | c | total |
|---|---|---|---|---|---|---|
| positive | 3 | 7 | 3 | 10 | 3 | 26 |
| negative | 4 | 5 | 9 | 9 | 8 | 35 |
| total | 7 | 12 | 12 | 19 | 11 | 61 |

**Table 5. A number of analyses with statistically significant results when using the default CONN denoising and the permutation-based approach.** Thresholded at t>2.3 and p<0.05 (corrected). Overall, 180 analyses were carried out, using 18 functional connectivity seeds for each personality domain and direction of effect.

| analysis | n | e | o | a | c | total |
|---|---|---|---|---|---|---|
| positive | 0 | 0 | 2 | 0 | 0 | 2 |
| negative | 2 | 1 | 0 | 3 | 2 | 8 |
| total | 2 | 1 | 2 | 3 | 2 | 10 |

state seed-to-voxel functional connectivity. Openness had a positive relationship with the temporal occipital fusiform cortex, the occipital fusiform gyrus, the lingual gyrus, the lateral occipital cortex (inferior division), the temporal fusiform cortex (posterior division), the lateral occipital cortex (superior division), and the parahippocampal gyrus (posterior division). A negative relationship between the seed-to-voxel functional connectivity of the default mode network and the other regions in the brain was found for neuroticism, extraversion, agreeableness, and conscientiousness. Neuroticism had a negative relationship with the occipital fusiform gyrus, the lateral occipital cortex (inferior division), the occipital pole, the lateral occipital cortex (superior division), the lingual gyrus, and the temporal occipital fusiform

cortex. Extraversion exhibited a negative relationship with the occipital fusiform gyrus, the lateral occipital cortex (inferior division), the lingual gyrus, the intracalcarine cortex, the lateral occipital cortex (superior division), the occipital pole, the cingulate gyrus (posterior division) and the precuneus cortex. Agreeableness was negatively related to the occipital pole, the lateral occipital cortex (superior division), the lingual gyrus, the lateral occipital cortex (inferior division), the occipital fusiform gyrus, the cuneal cortex, the intracalcarine cortex, the supracalcarine cortex, and the precuneus cortex. Conscientiousness had a negative relationship with the lingual gyrus, the occipital pole, the occipital fusiform gyrus, the lateral occipital cortex (superior and inferior division), the temporal occipital fusiform cortex, the cuneal cortex, and the intracalcarine cortex.

## Discussion

Adelstein, Shehzad, Mennes, DeYoung, Zuo, Kelly, et al. [22] showed a relationship between all five personality scales of the NEO-FFI and the inter-individual variations in the resting-state seed-to-voxel functional connectivity between the default mode network seeds and other regions of a brain. We endeavored to independently query the questions asked in the study of Adelstein's using a larger dataset with participants of similar age with a slightly higher proportion of males. We used the CONN toolbox with similar image preprocessing pipeline, as well as with an alternative denoising scheme and statistical method for multiple testing correction (Scripts can be found in S1 File). In general, we have not been able to reproduce the observed results, and our methodological analysis suggests that most if not all of the results reported by Adelstein, Shehzad, Mennes, DeYoung, Zuo, Kelly, et al. [22] constitute false positive findings.

Of course, the reproducibility of specific experimental results, including in neuroimaging research, is affected by many factors related to the experimental setup [29]. For instance, different MRI acquisition protocols and different preprocessing and denoising pipelines could have an effect on the observed results, for instance due to differential sensitivity to various imaging artifacts. Our acquisition protocol slightly differs from the one used by Adelstein, Shehzad, Mennes, DeYoung, Zuo, Kelly, et al. [22]. Compared to the authors, we have a lower sampling frequency of the functional images but a substantially longer acquisition. There is evidence that says that scan length and also sampling frequency can significantly affect the reliability of the functional connectivity measures from resting-state data [30].

On the other side, one should reasonably expect a substantial level of generalizability of results across experimental settings within the broad realm of 'standard' resting state fMRI, at least in qualitative terms. Of course, the appropriate sample size is crucial for obtaining robust results. In this sense, the use of only 37 subjects and multisession measurements in the original study undermines the generalizability of their results, and also complicates the use of the assumption of independent sampling at any statistical analysis at the group level.

The BOLD signal is generally more or less corrupted by various types of physiological artifacts or hardware-related artifacts like long-term instabilities of the scanner baseline [31]. There are two substantial differences between presented denoising strategies—mean-based intensity normalization and global signal regression. The global signal regression, based on a whole brain mask, is commonly used to reduce a physiological noise, such as respiratory and cardiac noise, under the assumption that the global signal is not correlated with task-induced signal [32]. Notably, the global signal regression was found to cause significant shifts in the functional connectivity values [25, 33, 34]. Additionally, the global signal is derived from the data itself and is an unknown mixture of neural and non-neural fluctuations and which affects the inter-regional correlations and complicate their interpretations [35]. But conclusions are rather controversial, and further investigation is needed.

Concerning the presented statistical methods, Gaussian random field-based cluster size tests are derived from a distribution approximation of cluster sizes based upon various parametric distributions [36]. Several assumptions like uniformly smoothed images and sufficiently high cluster-defining threshold are required [37–39]. For smoothness estimates for our first-level statistical maps, see S2 Table. However, there is an evidence suggesting that the gaussian random field-based tests tend to be less conservative under certain conditions compared to permutation tests [36, 40–42]; a strong evidence for the inflated false positive rate of gaussian random-field based tests was recently published by Eklund, Nichols, Knutsson [27]. Eklund, Nichols, Knutsson [27] also suggested a more stringent cluster-defining threshold (CDT) of z = 3.1, which had a better FWE control than the usual threshold of z = 2.3. To assess the impact of a more stringent threshold, we performed another analysis with CDT t>3.1 and the GRF approach (using both denoising strategies), which resulted in a substantial reduction of the number of significant analyses. Indeed, this approach showed smaller count of analyses with significant results (18 or 23 out of 180 tests, depending on denoising strategy). Importantly, the overlap between the t>3.1 results and those of Adelstein et al [22] dropped down to 8 (expected random overlap is 180 × (106 ÷ 180) × (18 ÷ 180) = 10.6) and 11 (expected random overlap is 180 × (106 ÷ 180) × (23 ÷ 180) = 13.54) for the original denoising and the default CONN denoising, respectively. In summary, when we use analysis with higher cluster-forming threshold, we obtain less significant results, but their overlap with the previous report [22] remains random. However, it still provided more positive results than permutation-based testing, which should be free of distributional assumptions (and provided 6 or 10 significant findings out of 180 tests, i.e. close to the 9/180 expected at p = 0.05 by chance). See S3 and S4 Tables for the number of significant analyses and S2 and S3 Figs for a summary visualization of obtained results.

Finally, if we wanted to formally test the observed positive count for the permutation test, we obtained 10/180 = 0.0556 with 95 percent confidence interval (0.027, 0.100) based on binomial distribution. This confidence interval safely includes 0.05, that is we can not disprove the null hypotheses. For positively dependent tests, this interval would be even wider, and there would need to be a really exotic shape of dependence between the tests for the interval not to include 0.05 and the null to be rejected. Finally, we do not formally test whether the observed positive count for the GRF tests is still in line with the null hypothesis—that could indeed happen for high positive dependence between the tests, but we already know from Eklund that a higher than nominal (>0.05) false positive rate is to be expected for GRF FWE procedure.

When comparing the results of permutation-based tests and GRF approach, it is evident that using permutation tests (that impose less distributional assumptions) always resulted in fewer clusters when compared to the Gaussian random field theory, irrespective of the denoising scheme used. But reasonably overlapping clusters between these statistical methods do exist, in particular, significant areas based on the permutation based inference are generally a restricted subset of the (less conservative) results based on GRF. Due to the number of analysis carried out (180 whole-brain regressions), we have limited the number of permutations to 5000.

For increased reproducibility, we share the first-level maps that are input for the analysis at https://osf.io/69kj8/, where both the first-level beta maps from both denoising options (for each subject, connectivity maps from each sead region) and the covariates (sex, age, and 5 personality dimension scores: neuroticism, extraversion, openness, agreeableness, conscientiousness).

## Conclusion

Our attempt at independent validation of the results by Adelstein, Shehzad, Mennes, DeYoung, Zuo, Kelly, et al. [22] was unsuccessful. While we have detected similarly extended

clusters of significant results across the whole brain as in the original study, the results had generally independent structure with respect to the original ones, in terms of space, pertinent seed regions, and personality dimensions. Reanalysis of the data using robust permutation-based correction for multiple testing problem yielded results consistent with the hypothesis of no relation between personality dimensions and resting state functional connectivity. While of course, this does not disprove the existence of such a link, it suggests that it may be much more subtle and elusive than it may seem at first sight.

## Supporting information

**S1 Fig.** Seed-based connectivity between 18 ROIs using the original (S1A Fig) and the default CONN denoising (S1B Fig).
(TIF)

**S2 Fig. Personality trait measures 'predicted' by rs-FC using the original denoising and the GRF approach.** Thresholded at t>3.1 and p<0.05 (corrected), positive—left, negative—right.
(TIF)

**S3 Fig. Personality trait measures 'predicted' by rs-FC using the default CONN denoising and the GRF approach.** Thresholded at t>3.1 and p<0.05 (corrected), positive—left, negative —right.
(TIF)

**S1 Table. MNI coordinates of seed ROIs.**
(PDF)

**S2 Table. Smoothness estimates for the first-level statistical maps.**
(PDF)

**S3 Table. Number of analyses, original denoising, thresholded at t>3.1, p<0.05 (corrected using GRF).**
(PDF)

**S4 Table. Number of analyses, CONN denoising, thresholded at t>3.1, p<0.05 (corrected using GRF).**
(PDF)

**S1 File. Preprocessing scripts.**
(ZIP)

## Author Contributions

**Conceptualization:** Jiří Horáček, Jiří Lukavský, Jaroslav Hlinka.

**Data curation:** David Tomeček, Renata Androvičová, Filip Děchtěrenko, Jan Rydlo, Jiří Lukavský, Jaroslav Hlinka.

**Formal analysis:** David Tomeček, Jaroslav Hlinka.

**Funding acquisition:** Jiří Horáček, Jiří Lukavský, Jaroslav Tintěra, Jaroslav Hlinka.

**Investigation:** David Tomeček, Renata Androvičová, Iveta Fajnerová, Filip Děchtěrenko, Jan Rydlo, Jiří Lukavský, Jaroslav Tintěra.

**Methodology:** Iveta Fajnerová, Filip Děchtěrenko, Jiří Horáček, Jiří Lukavský, Jaroslav Tintěra, Jaroslav Hlinka.

**Project administration:** Renata Androvičová, Jaroslav Hlinka.

**Software:** David Tomeček, Filip Děchtěrenko, Jan Rydlo.

**Supervision:** Jiří Horáček, Jaroslav Tintěra, Jaroslav Hlinka.

**Writing – original draft:** David Tomeček, Jiří Lukavský, Jaroslav Hlinka.

**Writing – review & editing:** David Tomeček, Jiří Horáček, Jaroslav Hlinka.

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
