## [Decision Letter · Decision Letter 0]

20 Feb 2020

PONE-D-19-26574

Personality reflection in the brain's intrinsic functional architecture remains elusive

PLOS ONE

Dear Hlinka,

Thank you for submitting your manuscript to PLOS ONE. After careful consideration, we feel that it has merit but does not fully meet PLOS ONE’s publication criteria as it currently stands. Therefore, we invite you to submit a revised version of the manuscript that addresses the points raised during the review process.

We would appreciate receiving your revised manuscript by Apr 03 2020 11:59PM. To enhance the reproducibility of your results, we recommend that if applicable you deposit your laboratory protocols in protocols.io, where a protocol can be assigned its own identifier (DOI) such that it can be cited independently in the future. For instructions see: http://journals.plos.org/plosone/s/submission-guidelines#loc-laboratory-protocols

We look forward to receiving your revised manuscript.

Kind regards,

Frantisek Sudzina

Academic Editor

PLOS ONE

Additional Editor Comments (if provided):

After receiving the two reviews, Dr. Milham was invited as a 3rd reviewer because of the journal policy regarding disputed articles; since he did not respond within a week, he was un-invited as any non-responding reviewer would be after 7 days.

Reviewers' comments:

Reviewer's Responses to Questions

**Comments to the Author**

1. Is the manuscript technically sound, and do the data support the conclusions?

Reviewer #1: Yes

Reviewer #2: Partly

2. Has the statistical analysis been performed appropriately and rigorously? 

Reviewer #1: Yes

Reviewer #2: Yes

3. Have the authors made all data underlying the findings in their manuscript fully available?

Reviewer #1: No

Reviewer #2: No

4. Is the manuscript presented in an intelligible fashion and written in standard English?

Reviewer #1: Yes

Reviewer #2: Yes

5. Review Comments to the Author

Reviewer #1: The work is well done and I recommend it for publication after completing the statistical results below

There should be reported:

- both corrected and uncorrected results of statistical tests

- measures of the smoothness of the data

Reviewer #2: Introduction is informative and covers well the background information.

The design of the study - a replication of the Adelstein 2011 study - is legitimate, more than doubling the sample size of the original study.

The results show that the cluster method used with a low defining cluster threshold report many detection of connectivity in relation to personality traits, but a more stringent statistical procedure (permutations) report far less results.

The discussion section addresses the points that required more in depth comments.

The take away message seems that the original Adelstein study should have been using a more stringent detection methods (eg permuations).

Questions:

- are the acquisition saggital ? could this have an effect in the connectivity study ?

- 180 analyses line 155: could the seed regions signals be correlated ? in that case, the bonferroni correction would be too conservative

- FSL’s cluster utility has been used : what was the cluster defining threshold: was it t=2.3? in that case, this has been shown to be a threshold too low (cf cluster failure Eklund PNAS article).

- The computations of the expected false detection rate assume independance of the tests - could you comment on this ?

- Could the dataset be released - ideally on openneuro - for replication ?

- Could the scripts be released - again for replication purposes ?

6. PLOS authors have the option to publish the peer review history of their article (what does this mean?). If published, this will include your full peer review and any attached files.

Reviewer #1: Yes: Vysata Oldrich M.D., PhD.

Reviewer #2: Yes: Jean-Baptiste Poline

---

## [Author Response · Author response to Decision Letter 0]

3 Apr 2020

Response to Reviewers

We thank the editor and the reviewers for expressing interest in the work and for the suggestions for improved presentation of our results. See below our detailed response including the description of detailed changes based on the comments.

Reviewer #1: The work is well done and I recommend it for publication after completing the statistical results below

There should be reported:

- both corrected and uncorrected results of statistical tests

- measures of the smoothness of the data

-> Thank you for the suggestions. 

Concerning inclusion of both corrected and uncorrected results of statistical tests, we have now included in the manuscript a link to a repository including detailed processed data and results, in particular also the 180 t-maps and the obtained mask of significant results when using uncorrected thresholding, GRF correction for multiple testing and permutation-based correction for multiple testing.

We also now include quantitative measures of the data smoothness in a new Supplementary Table refered to in the Discussion. 

Reviewer #2: Introduction is informative and covers well the background information.

The design of the study - a replication of the Adelstein 2011 study - is legitimate, more than doubling the sample size of the original study.

The results show that the cluster method used with a low defining cluster threshold report many detection of connectivity in relation to personality traits, but a more stringent statistical procedure (permutations) report far less results.

The discussion section addresses the points that required more in depth comments.

The take away message seems that the original Adelstein study should have been using a more stringent detection methods (eg permuations).

Questions:

- are the acquisition saggital ? could this have an effect in the connectivity study ?

-> We have acquired axial slices, which is the most common choice for EPI fMRI sequences and seems to be in line with the original study. In particular, while Adelstein et al. do not mention the slice direction, based on their reported slice count and image matrix, we are almost sure they used axial slices. 

- 180 analyses line 155: could the seed regions signals be correlated ? in that case, the bonferroni correction would be too conservative

-> Indeed, the seed region signals were of course correlated. We now comment on that in the manuscript and provide the subject-averaged correlation matrices in the Supplemental material. We agree that Bonferroni correction might be too conservative and approximating the degree of freedom is a very tricky issue given the complex design. Instead of such corrections, we advocate the use of permutation testing, or at least higher cluster-forming thresholds, as suggested by Eklund and commented by you below - we have included such analysis in the manuscript, see replies below.

- FSL’s cluster utility has been used : what was the cluster defining threshold: was it t=2.3? in that case, this has been shown to be a threshold too low (cf cluster failure Eklund PNAS article).

-> Yes, the original cluster-forming threshold in the paper by Adelstein et al. was t=2.3, as stated in the manuscript. We have replicated this analysis, but now also provide results of analysis with more conservative cluster-forming threshold t=3.1. This thresholding was shown to be less affected by the weaknesses of the GRF assumption (Eklund et al.), and indeed showed smaller count of significant analysis (18 or 23 out of 180 tests, depending on preprocessing). However, it still provided more positive results than permutation-based testing, which should be free of distributional assumptions (providing 6 or 10 significant findings out of 180 tests, i.e. close to the 9/180 expected at p=0.05 by chance). We have included the results and its discussion in the manuscript.

- The computations of the expected false detection rate assume independance of the tests - could you comment on this ?

-> Let us try to attempt to clarify the statistical subtleties, with apologies for a somewhat lengthy comment. Firstly, in the manuscript we do not quantify the false detection rate (which appears to be a less used term for false discovery rate, the ratio of true positives among all positive results of tests), neither at the level of a single analysis (across voxels), nor at the level of whole study (across the 180 hypotheses tested). What we do discuss is the observed rate of positive outcomes (among the 180 hypotheses tested), and compare it decriptively with the expectation on this quantity in case that all null hypotheses were true - the false positive rate, that is the expected value of the false positive ratio (i.e. the proportion of truly null hypotheses in which the null hypothesis is rejected based on testing). We conclude, that for GRF the count of tests is indeed much higher for t>2.3 than what would be expected by chance, but ascribe it to the known GRF problems (See Eklund). On the other side, for permutation-based test, the results are in line with the glocal null hypothesis. (6 or 10 instead of 9 expected positives out of 180 under null).

Secondly, in fact, the expected value of false positive rate (or generally any linear combination of random variables) does NOT assume independence of the tests (variables). In particular, such dependence between the tests does not affect the expected value of the count of positive results under null hypothesis (that the hypotheses are not true), it only affects the variance (distribution) of the positive (relative or absolute) result count around the expected value. In other words, under the family-wise null hypotheses the expected count of positive results of testing is 5% irrespective of dependence of tests, but in case the tests are dependent, higher variance around these 5% false positives is to be expected across replications of this study (if the tests were correct, which we know is not true for GRF, but should in principle be true for permutation-based testing). This variance will depent parametrically on the dependence between these tests, which is extremely cumbersome to be estimated (the dependance between the ROI time series is just the first step to assessing dependence between the tests themselves). 

Finally, if we wanted to formally test the observed positive count for the permutation test, we obtained 10/180=0.0556 with 95 percent confidence interval (0.027, 0.100) based on binomial distribution. This confidence interval safely includes 0.05, that is we can not disprove the null hypotheses. For positively dependent tests, this interval would be even wider, and there would need to be a really exotic shape of dependence between the tests for the interval not to include 0.05 and the null to be rejected. Finally, we do not formally test whether the observed positive count for the GRF tests is still in line with the null hypothesis - that could indeed happen for high positive dependence between the tests, but we already know from Eklund that a higher than nominal (>0.05) false positive rate is to be expected for GRF FWE procedure.

We now comment in a little more detail on these matters in the manuscript, however unless the editor/reviewers deem this useful for the reader to improve clarity, would prefer not to delve too deep into technical discussions in the manuscript.

- Could the dataset be released - ideally on openneuro - for replication ?

-> As mentioned in the reply tothe first reviewer, we now include a link to preprocessed data and results.

- Could the scripts be released - again for replication purposes ?

-> Yes, they are provided as supplementary material.

---

## [Editor Report · Decision Letter 1]

20 Apr 2020

Personality reflection in the brain's intrinsic functional architecture remains elusive

PONE-D-19-26574R1

Dear Dr. Hlinka,

We are pleased to inform you that your manuscript has been judged scientifically suitable for publication and will be formally accepted for publication once it complies with all outstanding technical requirements.

With kind regards,

Frantisek Sudzina

Academic Editor

PLOS ONE

---

## [Editor Report · Acceptance letter]

21 May 2020

PONE-D-19-26574R1 

Personality reflection in the brain's intrinsic functional architecture remains elusive 

Dear Dr. Hlinka:

I am pleased to inform you that your manuscript has been deemed suitable for publication in PLOS ONE. Congratulations! Your manuscript is now with our production department. 

With kind regards,

on behalf of

Dr. Frantisek Sudzina 

Academic Editor

PLOS ONE